# THE NEED FOR SPEED
# PRUNING TRANSFORMERS WITH ONE RECIPE

**Samir Khaki** [*]**, Konstantinos N. Plataniotis**
Department of Electrical and Computer Engineering
University of Toronto
Toronto, Canada
samir.khaki@mail.utoronto.ca

## ABSTRACT

We introduce the **O**ne-shot **P**runing **T**echnique for **I**nterchangeable **N**etworks (**OPTIN**) framework as a tool to increase the efficiency of pre-trained transformer architectures, across many domains, without requiring re-training. Recent works have explored improving transformer efficiency, however often incur computationally expensive re-training procedures or depend on architecture-specific characteristics, thus impeding practical wide-scale adoption across multiple modalities. To address these shortcomings, the OPTIN framework leverages intermediate feature distillation, capturing the long-range dependencies of model parameters (coined *trajectory*), to produce state-of-the-art results on natural language, image classification, transfer learning, and semantic segmentation tasks. Our motivation stems from the need for a generalizable model compression framework that scales well across different transformer architectures and applications. Given a FLOP constraint, the OPTIN framework will compress the network while maintaining competitive accuracy performance and improved throughput. Particularly, we show a $\leq 2\%$ accuracy degradation from NLP baselines and a $0.5\%$ improvement from state-of-the-art methods on image classification at competitive FLOPs reductions. We further demonstrate the generalization of tasks and architecture with comparative performance on Mask2Former for semantic segmentation and cnn-style networks. OPTIN presents one of the first one-shot efficient frameworks for compressing transformer architectures that generalizes well across *multiple* class domains, in particular: natural language and image-related tasks, *without re-training*. Code is available at: https://github.com/Skhaki18/optin-transformer-pruning.

## 1 INTRODUCTION

The inception of transformer architectures (Vaswani et al., 2017) marked the beginning of a new era in deep learning, since affecting various domains including natural language processing (Kenton & Toutanova, 2019), and vision-related tasks (Dosovitskiy et al., 2021). The transformers' straightforward design has enabled extensive applications to a variety of challenging problems, but it also brings a major drawback: high computational costs (Yu & Xiang, 2023). The computational resources required for training and inferencing with a transformer are often quite significant and pose a real impediment to wide-scale adoption, especially in resource-constrained environments, such as edge devices (Wang et al., 2020a). Recent works have proposed methods including quantization (Xiao et al., 2023), pruning (Ma et al., 2023), and knowledge distillation (Hao et al., 2022) to address this bottleneck, similarly explored in convolutional neural networks (CNN) compression (Li et al., 2017).

Despite much success in compressing CNNs, transformer architectures contain significant differences in their structure, often causing impediments for methods that work well in the former domain (Kwon et al., 2022; Yu & Xiang, 2023; Yang et al., 2023). Due to the massive size of Transformer models, some works have introduced various methods of compression, which can be loosely divided into *one-shot* and *iterative* (Zhang et al., 2022). *One-shot* methods generally consist of a pruning phase followed by *re-training* to recover the lost generalization performance, meanwhile, *iterative* processes

---

[*]Project Page: http://www.samirkhaki.com/optin-transformer-pruning/

can account for the training dynamics in model compression (Zhang et al., 2022). Unfortunately, in the past, both methods have often been limited to a particular architecture/task or required significant resources in the pruning and re-training processes. For user models that have already endured the expensive cost of training, there exists limited options for fast model compression (Kwon et al., 2022) that can be easily realized on standard hardware for different types of transformer architectures. The lack of a general approach to transformer pruning across multiple tasks and modalities provides sufficient motivation for the introduction of a unified framework; hence we introduce one of the first one-shot model compression techniques that generalize well over multiple tasks and architectures without incurring the cost of re-training.

In this work, we introduce the **O**ne-shot **P**runing **T**echnique for **I**nterchangeable **N**etworks (**OPTIN**) framework to efficiently compress modern transformers. The novelty is in its generalizability across domains and tasks, and its ability to produce competitive models without requiring re-training, thus enabling future application to larger models across many tasks. We apply OPTIN to natural language, and vision-related tasks, showing competitive performance with SoTA in these cases.

Our primary contribution rests on the ability of our OPTIN framework to produce transformers with competitive performance at reduced computational loads (FLOPs) across various task domains and architectures, that can be realized on standard hardware. In particular, we demonstrate superior performance on a variety of tasks in Language 4.1), Vision (Sec 4.2), and Application tasks (Sec 4.3), while maintaining competitive compression rates, without incurring the cost of re-training. Finally, we execute several extensive experiments from framework-specific settings to applications on transfer-learning and CNN networks to demonstrate OPTIN's robustness and generalizability over the task and architecture (Sec 3- 4.3).

## 2 RELATED WORKS

Due to the diversity of architectures and tasks discussed in our work, the following review of state-of-the-art methods provides an overview of efficient transformer design followed by recent developments in both language and vision domains.

**Domain Specific Design of Efficient Transformers** Transformers have enabled significant progress in the field of NLP (Vaswani et al., 2017; Kenton & Toutanova, 2019) and Computer Vision (Dosovit-skiy et al., 2021; Liu et al., 2021a). Efficiency improvements in transformers have stemmed from a variety of approaches including exploring hybrid architectures (Liu et al., 2021a), quantization techniques (Kim et al., 2021), knowledge distillation (Hao et al., 2022), and model pruning (Pan et al., 2021). Recently, TorchPruning (Fang et al., 2023) explored the application of multi-domain pruning by creating an inter-architecture dependency map, while UPop (Shi et al., 2023) introduced a unified pruning method for combined vision-language models. However, these methods have limitations, including architecture-specific dependencies and expensive re-training policies generally impeding wider-scale industry use (Fang et al., 2023). In contrast, our approach leverages intermediate feature distillation to compress pre-trained transformers in one shot across both language and vision-related tasks. Notably, our method operates effectively without re-training and scales well over a variety of complex architectures and task domains.

**Compressing Language Transformers** Several structured pruning methods have been introduced to compress models in the language domain. Attention Head pruning (Michel et al., 2019) explored the dynamics of attention heads across a transformer architecture to determine their individual impact on performance. Block-wise pruning (Lagunas et al., 2021) was motivated by removing block structures from weights under the movement pruning (Sanh et al., 2020) paradigm. DynaBERT (Hou et al., 2020) used distillation to transfer knowledge from a width-adaptive network onto a depth-adaptive smaller network. CoFi (Xia et al., 2022) explored the joint pruning between coarse and fine-grained modules in the transformer architecture. A recent work, namely Post-Training-Framework (PTF) (Kwon et al., 2022), was introduced to prune BERT in one-shot for NLP tasks, however, it leverages domain-related tricks to boost performance with a particular architecture and application. However, these methods have limitations, including dependence on architecture (Kwon et al., 2022; Lagunas et al., 2021; Hou et al., 2020) and expensive re-training procedures (Lagunas et al., 2021; Sanh et al., 2020; Hou et al., 2020; Xia et al., 2022). Focusing on the challenge of developing efficient transformers, we overcome these shortcomings by introducing a one-shot framework that produces competitive results at significant FLOPs reductions across several application domains.

**Compressing Vision Transformers** There have been several approaches to compressing vision transformers by focussing on different compute-intensive modules. $S^2$ViTE (Chen et al., 2021) explored structured sparsity by modifying first-order importance approximations enabling the dynamic sizing of attention heads in the ViT. SAViT (Chuanyang et al., 2022) developed a collaborative pruning scheme that analyzes component interaction to learn individual pruning ratios. Another stream of research introduced token reduction methods to accelerate both the training and inferencing throughput by gradually removing tokens from propagating forward in a Transformer (Kong et al., 2022; Fayyaz et al., 2022). EViT (Liang et al., 2022) builds on a Top-K approach by creating a fused token at each reduction stage to minimize the information lost from pruning. DynamicViT (Rao et al., 2021) introduced a lightweight prediction module to derive the importance scores of each patch per input. ToMe (Bolya et al., 2023) was introduced as one of the first one-shot methods in token reduction and leveraged bipartite matching to merge a fixed number of tokens at each transformer block regardless of input patches. TPS (Wei et al., 2023) furthered token reduction and merging, by identifying a pruned subset and squeezing the informative regions into a reserved subset of kept tokens. However, these methods still have limitations that prevent their widescale use including architecture specific design (Song et al., 2022; Chuanyang et al., 2022; Bolya et al., 2023) and expensive re-training policies (Chen et al., 2021; Chuanyang et al., 2022; Liang et al., 2022; Rao et al., 2021; Wei et al., 2023). In contrast, the OPTIN Framework leverages a one-shot approach to compress vision transformers across classification and semantic segmentation achieving competitive performance amongst state-of-the-art. The granularity and number of prunable components in the domain of vision transformers widely differ across state-of-the-art methods (Song et al., 2022). Similarly, the OPTIN Framework increases the base prunable components by allowing for the incorporation of token reduction methods through generating an optimal reduction policy as discussed in Sec. 4.2.

## 3 MEASURING TRAJECTORY

We aim to compress pre-trained transformer models by removing prunable parameters with minimal importance scores as determined by our salience metric without re-training. By analyzing the effects of parameter removal on deeper layers in the network, our *trajectory* metric is able to better select important parameters by leveraging long-term inter-layer dependencies in the model.

**Problem Statement.** Given a model $f$ (with $N$ layers) expressed by its collection of weights $[\theta_0, \theta_1, \cdots \theta_N] \in \mathbb{R}^{N \times d}$, a pruned subset has weight collection $[\theta'_0, \theta'_1, \cdots \theta'_N] \in \mathbb{R}^{N \times d}$, such that $\theta' = m \odot \theta$ where $m \in \{0,1\}^d$ is a binary mask and $\odot$ is the element wise product operator. We define this pruned subset to be optimal if it satisfies the cost constraint and results in the minimum decrease in validation error from the base model expressed with $\mathcal{L}_{err}$. Formally, we express this as:

$$\text{argmin}_{[\theta'_0, \theta'_1, \cdots \theta'_N]} \quad \mathcal{L}_{err}(f(X, [\theta_0, \theta_1, \cdots \theta_N]), f(X, [\theta'_0, \theta'_1, \cdots \theta'_N]))$$
$$\text{subject to} \quad \mathcal{C}([\theta'_0, \theta'_1, \cdots \theta'_N]) \leq C. \tag{1}$$

where, the optimal selection of weights $[\theta'_0, \theta'_1, \cdots \theta'_N]$ meets the cost requirement $C$ while retaining the minimum drop in validation performance on the dataset $X$.

**Approach** In general, for each transformer block we define the prunable weights as the collection of attention heads and fully connected neurons, individually denoted by $\theta_{i,j}$ where the parameter is located in layer $i$ at an arbitrary index $j$. The exact prunable components for each task are described in Appendix A.8. We progressively mask each prunable parameter, and compute the importance score by executing a forward pass that originates from layer $i$ and propagates forward to the logit prediction. In particular we express the masking of weight $j$ in layer $i$ as $MASK_j \odot \theta_i$ where $MASK$ is the instance of $m$ with a single zero at location $j$, as used in Algorithm 1. This masked forward pass yields subsequent layer-wise activations and output logits, which are both used in computing the trajectory of parameter, $\theta_{i,j}$. We denote the cumulative importance of parameter, $\theta_{i,j}$, as $\mathcal{I}_{i,j}$. Referring to the optimization problem in Eq. 1, we use our importance metric as a proxy for determining which parameters will least affect the validation error, $\mathcal{L}_{err}$, on the testing dataset, $X$. Upon computing all importance scores $\mathcal{I}_{i,j}$, we employ an expedited mask-search policy, from (Kwon et al., 2022), which computes the optimal configuration in a faster polynomial-time by sequentially adding parameters in descending importance. Further details on the search method are discussed in Appendix A.1. We introduce the OPTIN Framework algorithm in Algorithm 1 and Diagram in Fig. 1.

**Algorithm 1** OPTIN Framework for Model Compression

1: **Inputs:** FLOPs Constraint ($\mathcal{C}$), Importance Scores ($\mathcal{I} \leftarrow []$), model, batch
2: $([\mathcal{F}_0, \cdots \mathcal{F}_N], logits) \leftarrow model(batch)$          ▷ Pre-Compute Forward Pass
3: **for** $\theta_i$ in $[\theta_0, \theta_1, \cdots \theta_n]$ **do**          ▷ layer: $i$
4:      **for** $j \in$ range($d$) **do**          ▷ weight: $j$
5:          model[$i$].weight $\leftarrow \theta_i * MASK_j$          ▷ Apply Mask to Weight $(i,j)$
6:          $([\mathcal{F}'_0, \cdots \mathcal{F}'_N], logits') \leftarrow$ model($batch$)          ▷ Compute Masked Forward Pass
7:          $\mathcal{I}_{i,j} \leftarrow \sum_{z=i+1}^{N} \mathcal{L}_{MD}(F'_z, F_z) + \lambda \mathcal{L}_{KD}$          ▷ Apply Eq.2 and 3
8:      **end for**
9: **end for**
10: Reduced Model $\leftarrow$ SEARCH($\mathcal{I}, \mathcal{C}$)

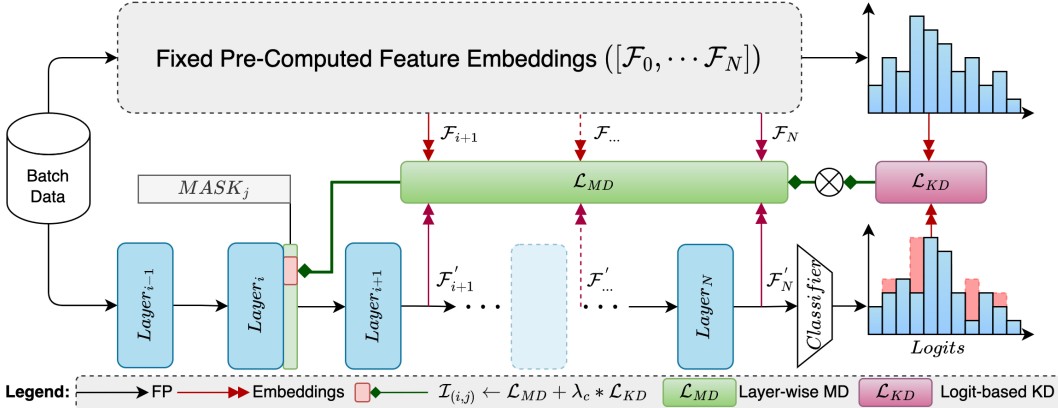

Figure 1: Illustrates the computation of the OPTIN Frameworks trajectory metric on weight $\theta_{i,j}$. By applying a mask to weight $\theta_{i,j}$ in Layer$_i$ and executing a forward pass, the OPTIN framework can measure the effect on future layer embeddings (*trajectory*), as an indicator of weight importance. $\mathcal{L}_{MD}$ is the manifold distillation loss computed between layer embeddings at each transformer block, while $\mathcal{L}_{KD}$ is the KL-Divergence computed between the original logits and those due to the masked weight. The combination losses are further detailed in the Weight Importance heading under Sec. 3

**Parameter Importance** Prior to assigning an importance score to each parameter, we define what it means to be "important". While many prior works have coined the importance of parameters by analyzing their intrinsic structure and error dynamics (Kurtic et al., 2022), these are not necessarily the most intuitive. For instance, magnitude-based metrics (Li et al., 2017) capture the intrinsic dominant property of individual weight structures, however, fail to capture their interactions with the data, meanwhile, activation methods (Lin et al., 2023) can capture in-place reconstruction errors, however, they may obscure the global impact on deeper layers in the model. Motivated by capturing the long-term effects of weights, we frame the problem as identifying which weights are more important based on how much they affect subsequent layer embeddings, hence we coin the measure *trajectory*.

**Effect on Trajectory** To compute the trajectory of a weight, $\theta_{i,j}$ we follow a 2-step procedure. Firstly, we measure layer-wise activation errors prior to the LayerNorm operator at each block subsequent to the layer of interest. We conducted an ablative study in 1a showing the effect of using pre-layer norm embeddings. We first define the feature output of layer $i$ as $\mathcal{F}_i \in \mathbf{R}^{B \times T \times D}$, where $B$ is the batch size, $T$ is the token length and $D$ is the embedding dimension. We similarly express the feature output of the masked network using the prime$'$ symbol. Inspired by distillations works (Sajedi et al., 2023a; Peng et al., 2019), we compute the layer-wise error by adopting fine-grained manifold distillation (Hao et al., 2022). A reshaping operator $\psi(\cdot) \in \mathbf{R}^{BT \times D}$ leverages patch and batch level information, defining the relational map and associated metric as:

$$\mathcal{M}(F_i) = \psi(F_i)\psi(F_i)^T \qquad \mathcal{L}_{MD}(F'_i, F_i) = ||\mathcal{M}(F'_i) - \mathcal{M}(F_i)||^2_F \qquad (2)$$

However, unlike previous works, we do not use this loss to guide training or distillation, rather, we express it as an in-place metric to help understand parameter importance throughout the network.

If the masked weight is at position $j$ in layer $i$, the computed error is accumulated over both the dimension $D$ and at each layer $l$ in the range $[i+1, N]$. The choice of error aggregation was explored in Tab. 1c and clearly demonstrated the benefit of the sum operator. Additionally, we explored the

| Dataset | Emb. | Acc. |
|---|---|---|
| MNLI | L-Norm | **81.92** |
| MNLI | FFN | **81.90** |
| MNLI | IM-Dense | 81.83 |
| | | |
| ImageNet | L-Norm | **71.27** |
| ImageNet | FFN | **71.25** |
| ImageNet | IM-Dense | 70.90 |

(a) **Embedding Choice.** The dense output layer best informs weight performance when compared layer-wise.

| Dataset | Temp. | Acc. |
|---|---|---|
| MNLI | 1 | 82.01 |
| MNLI | 2 | 82.11 |
| MNLI | **4** | **81.90** |
| MNLI | 8 | **82.14** |
| ImageNet | 1 | 70.54 |
| ImageNet | 2 | 70.77 |
| ImageNet | **4** | **71.25** |
| ImageNet | 8 | 71.00 |

(b) **Temperature Choice.** The choice of $T = 4$ best captures the effect on the logits by removing weights.

| Dataset | Aggregate | Acc. |
|---|---|---|
| MNLI | sum | **81.90** |
| MNLI | mean | 80.75 |
| | | |
| ImageNet | sum | **71.25** |
| ImageNet | mean | 71.15 |

(c) **Layer Error Aggregation.** The cumulative error over the layer's manifold distribution best captures weight importance.

| | Dataset | $\mathcal{L}_{MD}$ | $\mathcal{L}_{KD}$ | $\lambda_c$ | Acc. |
|---|---|---|---|---|---|
| | MNLI | ✓ | - | - | 81.71 |
| | MNLI | - | ✓ | - | 80.91 |
| Language | MNLI | ✓ | ✓ | 10 | 81.74 |
| | MNLI | ✓ | ✓ | 1 | 81.86 |
| | MNLI | ✓ | ✓ | 0.1 | **81.90** |
| | MNLI | ✓ | ✓ | 0.01 | **82.12** |
| | ImageNet | ✓ | - | - | 70.34 |
| | ImageNet | - | ✓ | - | 68.85 |
| Vision | ImageNet | ✓ | ✓ | 10 | 70.82 |
| | ImageNet | ✓ | ✓ | 1 | 70.85 |
| | ImageNet | ✓ | ✓ | 0.1 | 70.99 |
| | ImageNet | ✓ | ✓ | 0.01 | **71.25** |

| Dataset | Type† | Acc. |
|---|---|---|
| MNLI | $[i]$ | 78.89 |
| MNLI | $[i+1]$ | 80.07 |
| MNLI | $[i, N]$ | 81.65 |
| MNLI | $[i+1, N]$ | **81.90** |
| ImageNet | $[i]$ | 68.91 |
| ImageNet | $[i+1]$ | 69.55 |
| ImageNet | $[i, N]$ | 70.04 |
| ImageNet | $[i+1, N]$ | **71.25** |

(d) **Component Analysis** Combining both $\mathcal{L}_{MD}$ and $\mathcal{L}KD$ captures the best information (see Eq. 3). Based on the value of $\lambda$, we can see $\mathcal{L}_{MD}$ should have a stronger weight in parameter selection. Further details on the hyperparameter choice are in Appendix A.2.

(e) **Layer Trajectory Depth.** Accumulating $\mathcal{L}_{MD}$ over deeper layers performs best. † indicates which layer(s) to measure $\mathcal{L}_{MD}$, relative to current layer $i$ and final layer $N$.

Table 1: **Ablative Experiments on Trajectory** using BERT$_{BASE}$ (Kenton & Toutanova, 2019) on the GLUE benchmark MNLI dataset (Wang et al., 2019), and DeiT-Ti (Touvron et al., 2021) on the ImageNet-1K dataset (Deng et al., 2009) to explore the effect parameters on model performance. We measure one-shot post-pruning accuracies over various configurations on both the language and vision datasets. In particular **(a)** explores locations to extract features for distillation loss: L-Norm (After Layer Normalization), FFN (After Dense output layer), IM-Dense(After Dense Embedding Layer). **(b)** examines the effect of temperature in the KL-Divergence Formulation, **(c)** explores the effect of summing or averaging over the layer distillation error, **(d)** explores the effect of metrics $\mathcal{L}_{MD}$ and $\mathcal{L}_{KD}$ as well as their balancing paramter $\lambda$, **(e)** explores which layer to accumulate the distillation error in relation current layer $i$. Compression rates remain consistent with Tab. 7 and Tab 3. The baseline performance of BERT$_{BASE}$ on MNLI is **84.53%**, meanwhile DeiT-Ti on ImageNet-1K is **72.20%**%. Our default settings are marked in green.

effect of modifying which layers were relevant to the trajectory – see Tab. 1e – overall it was evident that using subsequent layers yielded the best result, correctly aligning with the original motivation.

**Effect on Logits** Next we compute the effects on the logit prediction as shown in Fig. 1 with $\mathcal{L}_{KD}$. Several works have shown the effects of using logit predictions to guide the training process with distillation (Hao et al., 2022; Zhao et al., 2022) or correlation (Sajedi et al., 2024; 2023b). However, in this work, we use the $\mathcal{L}_{KD}$ loss (defined in (Hinton et al., 2015)) as an in-place metric to quantify the importance of a particular weight. We ablate the temperature value in Tab. 1b. Thus, if masking a particular weight produces a larger $\mathcal{L}_{KD}$, we would hypothesize that it is a more important weight.

We can formalize the importance of a particular weight $j$ at layer $i$ with iterator $z$ as:

$$\mathcal{I}_{i,j} = \sum_{z=i+1}^{N} \mathcal{L}_{MD}(F_z', F_z) + \lambda \mathcal{L}_{KD} \tag{3}$$

where $\mathcal{I}_{i,j}$ is defined for a particular weight in a particular layer, $\lambda$ controls the contribution effect of KD, and $\mathcal{L}_{MD}$ compares the pruned and original resulting embeddings in deeper layers. We ablate the effect of different $\lambda$ values as well as the contribution of each loss individually in Tab. 1d. Further, we show that applying a greater importance on $\mathcal{L}_{MD}$ results in better parameter selection. Further details on the contribution hyperparameter are expressed in Appendix A.2. We also provide the algorithm for applying OPTIN on a generic model instance in Algorithm 1.

## 4 EXPERIMENTAL DESIGN

In this section, we demonstrate the effectiveness of the OPTIN Framework, at improving model performance and throughput given strict FLOP reduction ratios. We introduce implementation and evaluation details to ensure reproducibility and benchmark our method with state of art in natural language and image classification to illustrate the potential of our one-shot framework. We further investigate the applications in transfer learning, alternate architectures, and downstream tasks to show the generalizability of our method across tasks and architectures.

**Experimental Setup** We implement our method using transformers from the HuggingFace Library (Wolf et al., 2020) and infrastructure from PyTorch (Paszke et al., 2019). The majority of our experiments explore using the **OPTIN** Framework to improve *off-the-shelf* models without re-training. The exceptions include select experiments in the Applications Section, see Sec. 4.3. The OPTIN Framework computes parameter importance on the basis of training data in a gradient-free forward pass. The amount (batch) of data used to compute the scores is ablated in Appendix A.6. Finally, details on the prunable components under each setting are described in Appendix A.8.

**Datasets & Networks** The OPTIN Framework is tested against a variety of network architectures and datasets to ensure generalizability over both the task and model domains. For Natural Language Processing, OPTIN is evaluated on the GLUE Benchmark (Wang et al., 2019) using the $BERT_{BASE}$ (Kenton & Toutanova, 2019) architecture. For Image Classification, both ImageNet1-K (Deng et al., 2009) and CIFAR10 (Krizhevsky et al., 2009) were used with the DeiT-Ti/S/B (Touvron et al., 2021), ViT-B (Dosovitskiy et al., 2021), and a VGGNet (Simonyan & Zisserman, 2014) architecture to demonstrate the OPTIN Framework's robustness on model type/size, image datasets, and transfer learning. For Semantic Segmentation, the Cityscapes Dataset (Cordts et al., 2016) was used with the Mask2Former (Cheng et al., 2022) with a Swin-Ti backbone (Liu et al., 2021a) to show how the OPTIN Framework could be used to maintain competitive performance and throughput at constrained FLOPs. Further details on dataset selection are in Appendix A.3

**Evaluation Metrics** With the goal of model compression, we evaluate models based on their accuracy (or mIoU for segmentation) given a FLOP reduction. Details regarding the metric choice for the corresponding task can be found in Appendix A.4. In select cases, we include either a measurement of throughput speed (images/data per second) or latency expressed as a ratio of the improved inferencing speed to that of the baseline. All time measurements are captured over 300 iterations on an Nvidia RTX 2080 using a 100-iteration warmup.

### 4.1 LANGUAGE EXPERIMENTS

**Performance on NLP Benchmarks** In Tab. 7 we investigate the OPTIN Framework for compressing language models on the GLUE dataset using $BERT_{BASE}$. Measuring performance and throughput speeds, we show a relatively low average decline in baseline accuracy ($\leq 2\%$) at a $40\%$ FLOPS compression rate. Similarly, we benchmark our performance with a leading one-shot SoTA method: Post-Training-Pruning-Framework (PTF)(Kwon et al., 2022) at the same compression rate and show superior performance. In particular, we compare with the *mask search* results from PTF, as subsequent phases in their method could be stacked on other post-training pruning methods (refer to Appendix A.7 for exetended comparisons). We demonstrate robustness over various compression ratios in Fig. 2 where we benchmark OPTIN against pipelines that incorporate re-training, including CoFi (Xia et al., 2022), DynaBert(Hou et al., 2020), SLIP (Lin et al., 2020b), EBERT(Liu et al., 2021b), BMP (Lagunas et al., 2021) and FLOP (Wang et al., 2020b). Despite the added re-training phase in other methods, the OPTIN Framework is able to retain competitive test performance over a variety of compression ratios thus establishing a compelling argument for retraining-free pipelines.

### 4.2 VISION EXPERIMENTS

**Extending to Image Classification** Transitioning the OPTIN Framework from the language domain to the vision domain, we were required to increase the number of prunable components. Comparable

| Method | MNLI | QQP | QNLI | SST | STS-B | MRPC |
|---|---|---|---|---|---|---|
| BERT$_{BASE}$ | 84.53 | 91.00 | 91.41 | 93.57 | 88.90 | 86.27 |
| PTF[†] | 81.21 | 89.99 | 88.38 | 92.13 | 87.10 | 83.14 |
| **OPTIN[‡]** | **81.90** | **90.06** | **88.49** | **92.24** | **87.25** | **85.13** |

Table 2: **Natural Language Benchmarks.** Comparing OPTIN performance on the GLUE (Wang et al., 2019) benchmark (refer to A.7 for additional results). The relative FLOP constraint is set to 60% for a fair comparison.

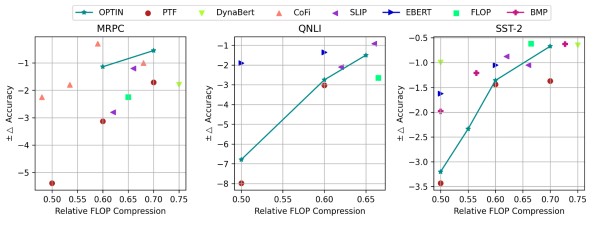 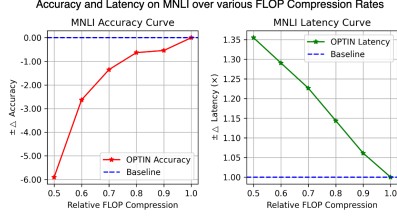

Figure 2: **Natural Language FLOPs vs Accuracy.** We directly benchmark the OPTIN Framework against leading state-of-the-art methods in natural language model compression. Due to the numerous different baselines reported in each work, we plot the relative performance drops for each method. On the right, we compare the performance gap with latency showing that with an average drop of $\leq 1.75\%$ we can achieve a $1.25\times$ speedup in throughput purely from static model size reduction.

works have used a larger number of components including the pruning of Q-K-V layers in each attention head, tokens & patches, and final embeddings in each transformer block (Zhu et al., 2021; Wei et al., 2023; Pan et al., 2021). Thus the state of the art in the field of transformer pruning widely differs in the granularity and consistency of the pruned components. With a priority on reducing real-world inference time, we extend OPTIN to include a variant of token reduction; a similar adaptation was made in CP-ViT (Song et al., 2022). In particular, we derive a modified *trajectory* formulation to rank tokens between each transformer block as described in Appendix A.9. By incorporating the *trajectory* metric for layerwise-token ranking, the OPTIN framework can deduce the optimal number of tokens to preserve between each transformer block, and can thus create an informative token reduction schedule that can be leveraged by any token reduction method. In particular, we were inspired by a recent work, ToMe (Bolya et al., 2023) which features an efficient token merging technique based on bipartite matching that removes tokens between transformer blocks either at a constant or linearly decreasing schedule. We incorporate ToMe as a method of merging tokens based on the optimal number of reduced tokens per layer determined by the OPTIN framework search. Since our framework produces the reduction schedule, we can leverage the benefit of batching as the number of tokens per image will be constant, and the methods of merging or reducing can be selected by any user – we ablate the bipartite matching scheme with a random pruning scheme in Appendix A.6 and show similar improvements when using the OPTIN Framework.

**Image Classification Results** In Tab. 3 we benchmark our proposed re-training free method on the ImageNet-1K dataset with the baseline, and SOTA results to show competitive performance at given FLOPs reductions. We offer two configurations: $\beta$ (base) denotes the base OPTIN Framework without the additional prunable components (directly shifted from the language domain), $\tau$ (expanded) denotes the incorporation of token reduction into our search space. Compared with methods that perform re-training, the OPTIN Framework produces competitive performance, particularly with a $0.5\%$improvement at a $5\%$ lower FLOPs with respect to SAViT(Chuanyang et al., 2022). Comparing with methods that have removed re-training, we note that VTP (Zhu et al., 2021) still includes additional sparsity-regularization training, PoWER (Goyal et al., 2020) still includes the auxiliary network training with *soft-extract*, and HVT (Pan et al., 2021) still reduces FLOPs via re-training the architecture with a pooling method. However our method is considered a fundamentally one-shot design and despite lacking these additional pruning artifacts & components, is still able to outperform the current SoTA, with the best result on DeiT-Small outperforming CP-ViT(Song et al., 2022) by $0.4\%$ at a $\sim 10\%$ higher FLOPs reduction. To further benchmark our performance in perspective of a wider FLOPs spectrum and more model compression methods, we introduce Fig 3 which includes: X-Pruner (Yu & Xiang, 2023), WDPruning (Yu et al., 2022a), S2VITE/SSP(Chen et al., 2021), SCOP (Tang et al., 2020), HVT (Pan et al., 2021), SAViT(Chuanyang et al., 2022), VTP (Zhu et al., 2021), PoWER(Goyal et al., 2020), CP-ViT(Song et al., 2022) and UVC (Yu et al., 2022b). Despite our lack of re-training, the OPTIN framework produces competitive results over various flop ratios.

| Method | DeiT Tiny | | DeiT Small | |
|---|---|---|---|---|
| | FLOPs(G) | Acc(%) | FLOPs(G) | Acc(%) |
| Baseline | 1.3 | 72.2 | 4.6 | 79.8 |
| Re-Trained | | | | |
| SSP | $0.99^{\downarrow 23.7\%}$ | 68.59 | $3.15^{\downarrow 31.6\%}$ | 77.74 |
| $S^2$ViTE | $0.99^{\downarrow 23.7\%}$ | 70.12 | $3.15^{\downarrow 31.6\%}$ | 79.22 |
| SAViT[††] | $0.98^{\downarrow 24.4\%}$ | 70.72 | - | - |
| Not Re-Trained | | | | |
| VTP[†] | $1.00^{\downarrow 21.7\%}$ | 69.37 | $3.65^{\downarrow 20.7\%}$ | 77.35 |
| PoWER[†] | $1.02^{\downarrow 20.3\%}$ | 69.56 | $3.61^{\downarrow 21.5\%}$ | 77.02 |
| HVT[†] | $1.01^{\downarrow 21.2\%}$ | 68.43 | $3.66^{\downarrow 20.5\%}$ | 76.72 |
| CP-ViT[†] | $1.00^{\downarrow 23.0\%}$ | 71.06 | $3.64^{\downarrow 21.0\%}$ | 78.84 |
| **OPTIN**$_\beta$ | $1.1^{\downarrow 15.46\%}$ | 67.51 | $4.11^{\downarrow 11.2\%}$ | 77.01 |
| **OPTIN**$_\tau$ | $0.91^{\downarrow 29.7\%}$ | **71.25** | $3.15^{\downarrow 31.6\%}$ | **79.24** |

Table 3: **Pruning ImageNet-1K.** Benchmarking the performance of OPTIN using DeiT-Tiny/Small. [†] methods are reproduced in (Song et al., 2022) without re-training. [††] DeiT-S result from (Chuanyang et al., 2022) is excluded as it performs superior to the available baseline. OPTIN framework runs without re-training producing both the $\beta$ and $\tau$ configurations.

| Model | Method | ImageNet-1K | | Transfer→C-10 | |
|---|---|---|---|---|---|
| | | FLOPs(G) | Acc(%) | FLOPs(G) | Acc(%) |
| DeiT-S | Baseline | 4.6 | 79.8 | 4.6 | 97.13 |
| | **OPTIN**$_\tau$ | $3.52^{\downarrow 23.7\%}$ | **79.01** | $2.30^{\downarrow 50.0\%}$ | **96.60** |
| ViT-B | Baseline | 17.47 | 75.40 | 17.47 | 98.01 |
| | **OPTIN**$_\tau$ | $13.33^{\downarrow 23.7\%}$ | **72.98** | $8.77^{\downarrow 50.0\%}$ | **97.82** |

Table 4: **Transfer Learning on CIFAR Dataset.** Benchmarking the performance of OPTIN on the CIFAR-10 Datasets. Models were pre-trained on ImageNet-1K, pruned through the OPTIN Framework $\tau$ configuration, and transferred learned at a more aggressive pruning rate onto the CIFAR-10 (C-10) Dataset.

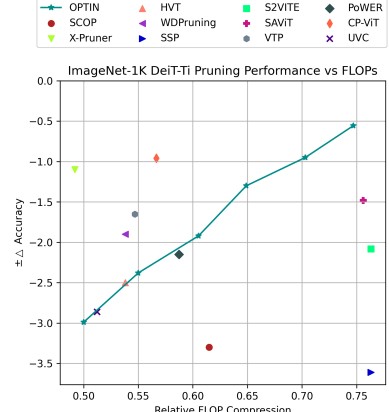

Figure 3: **DeiT-Ti FLOPs vs Accuracy** Benchmarking OPTIN$_\tau$ over a range of FLOP reductions on ImageNet-1K. The OPTIN Framework shows strong robustness over various FLOP constraints without re-training.

| Method | FLOPs(G) | $\pm\triangle$Acc(%) |
|---|---|---|
| ViT-B | 17.47 | – |
| ToMe | 11.50 | $\downarrow 1.88$ |
| **OPTIN**$_{\tau(\infty)}$ | **11.45** | $\downarrow$ **0.71** |
| DeiT-B | 17.6 | – |
| Dyn-ViT[†] | 11.81 | $\downarrow 1.17$ |
| Top-K[†] | 11.81 | $\downarrow 0.94$ |
| EViT[†] | 11.81 | $\downarrow 0.86$ |
| ToMe[†] | 11.81 | $\downarrow 0.80$ |
| TPS[††] | 11.51 | $\downarrow 0.71$ |
| **OPTIN**$_{\tau(\infty)}$ | **11.75** | $\downarrow$ **0.52** |

Table 5: **Token Reduction** Benchmarking OPTIN$_{\tau(\infty)}$ configuration. [†] & [††] detailed in the main text.

For completeness, we chose to introduce a third configuration $\tau_{(\infty)}$ which only applies OPTIN to creating token reduction schedule, while leveraging ToMe for merging. Under this constraint, we evaluate our method with state-of-the-art token reduction methods: including DynamicViT (Dyn-ViT) (Rao et al., 2021), Top-K (Haurum et al., 2023), EViT (Liang et al., 2022) and TPS (Wei et al., 2023) in Tab 5 and show superior performance under our framework without re-training. † methods follow setup & produced results in (Haurum et al., 2023), we convert a token percentage to FLOPs reduction to benchmark our method.†† estimated from (Wei et al., 2023). We further complement this with a more detailed comparison against the schedules using constant and linearly decreasing reduction schedules in ToMe over a wide variety of FLOP constraints in Appendix A.6. Ultimately this provides a compelling case for OPTIN's ability to effectively determine average token importance in transformer architectures.

**Transfer-Learning for Image Classification** To demonstrate the transferability of our compressed models, we obtain the pruned networks from ImageNet-1K and apply transfer learning to the CIFAR-10 dataset. We choose to include DeiT-S and ViT-B for model size diversity. Benchmarking against baseline models, in Tab 4 we show significant recovery of performance when transferring learning onto CIFAR-10 at extensive FLOPs reduction ratios. Although we show re-training is not necessary when it comes to pruning on a specific dataset & task, we evidently show that the method works well under the transfer learning paradigm for different downstream purposes.

### 4.3 APPLICATIONS
We further explore downstream tasks and architectures that can similarly benefit from the OPTIN Framework. Particularly, high-resolution (HR) semantic segmentation is a compute-intensive task,

and we explore how OPTIN maintains competitive performance and increases throughput speeds in Tab 4a and Fig 4b. Finally, we explore brief experiments on CNN pruning to show generalizability beyond standard transformer modules.

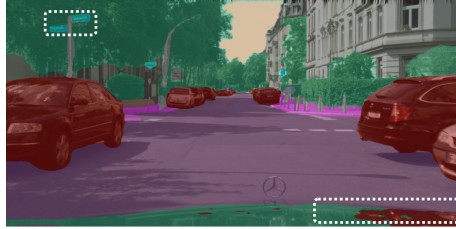

| Method | FLOPs(M) | Top-1(%) | Epochs |
|--------|----------|----------|--------|
| Baseline | 313.73 | 93.96 | - |
| $L^1$ | 206.00 | 93.40 | - |
| HRank | 131.17 | 93.73 | 200-300 |
| CFDP | 131.17 | **94.10** | 200-300 |
| **OPTIN** | 131.17 | **94.10** | **100-150** |

Figure 4: **Pruning on CNN.** Benchmarking the performance of OPTIN on the CIFAR-10(Krizhevsky et al., 2009) dataset using VGG-16-BN. OPTIN outperforms previous model compression techniques.

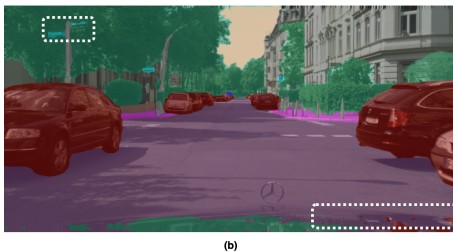

| Method | FLOPs↓ | Params↓ | mIoU(%) | Latency(↓) |
|--------|--------|---------|---------|------------|
| Baseline | - | - | 78.81 | - |
| **OPTIN$_\beta$** | 24.2% | 46.6% | 74.57 | 13% |

(a) **Mask2Former (Swin-Ti): Semantic Segmentation.** Benchmarking OPTIN$_\beta$ on the CityScapes (Cordts et al., 2016) FLOPs, Params., and Latency reduction is measured relative to the SWIN encoder.

(b) **High-Resolution Segmentation** (a) Baseline Mask2Former; (b) OPTIN Framework at a FLOPs reduction of $\sim 25\%$. The minimal observable discrepancy is encircled in white rectangles.

Figure 5: Evaluated OPTIN$_\beta$ on HR (1024x2048) Segmentation (**(a)** Quantitative; **(b)** Qualitative).

**Exploring Semantic Segmentaion** To demonstrate the OPTIN framework's generalizability to complex architectures and downstream tasks, we apply model compression to the Mask2Former Architecture with the Swin-Tiny backbone on the Cityscapes dataset. We specify the selected prunable components in Appendix A.8. In Tab. 4a we show impressive performance despite roughly a 24% reduction in FLOPs and 47% reduction in parameters of the endocer. Qualitatively we can see a strong resemblance between the original and compressed network, with a small discrepancy in predictions towards the bottom right of the frame in an already difficult-to-segment region (as evidenced by the unclear segmentation in the original prediction) and on the traffic sign towards the top left in Fig. 4b.

**Exploring CNN Architectures** To demonstrate the potential applications of OPTIN onto CNN architectures, we extend our trajectory measure as described in Appendix A.10. In Tab. 4a we compare the model compressed through the OPTIN Framework with the baseline on the VGG-16-BN architecture, a heuristic approach ($\mathcal{L}^1$) (Li et al., 2017) and two leading state-of-ther-art: HRank (Lin et al., 2020a) and CFDP(Khaki & Luo, 2023). Following previous works (Lin et al., 2020a), fine-tuning has been shown to be required post-compression. However as evident in Tab 4a, following the same training procedures as HRank, we were able to outperform all methods at a much faster convergence speed given comparable FLOPs reductions.

## 5 CONCLUSION

In this work, we introduced OPTIN as a one-shot technique to enable efficient compression of modern pre-trained transformer architectures *without* re-training. OPTIN exploits the trajectory of prunable components in the transformer architecture to enable a smarter criterion for parameter selection. In this work, we've explored several domains including natural language processing, image classification, transfer learning and semantic segmentation tasks. We additionally show how our method can work in concert with existing token reduction modules to produce even stronger competitive results in the image domain. We further expanded our method to show robustness on prior CNN-style architectures opening future avenues of research into fused architectures. In all cases, we've shown robustness against compression rates and competitive performance including against methods that perform re-training. We complement our performance improvements with synonymous improvements in throughput speed enabling the practical use of our framework. In the future, we plan to explore more complex architectures and tasks in addition to expanding the number of prunable components to further the cause in an efficient design of transformer models.

## 6 REPRODUCIBILITY STATEMENT

The attached supplemental code contains a framework with the algorithms and metrics behind our main results. All of our adapted $\mathcal{L}$ formulations are described in the main paper: Sec 3 or Appendix A.9, A.10, and are implemented in the supplemental code. By our innate one-shot structure, there are no training augmentations applied as we don't re-train. The exception is for transfer learning and re-training on the CNN architectures. For these, we adopt the standard augmentations from HRank (Lin et al., 2020a). Our datasets and evaluation metrics are described in Appendix A.3, A.4.

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

# A    APPENDIX

The appendix is structured to provide details that matches elicitation from the main text. Experiments and discussions included in the appendix serve as supplemental information to provide a greater context to claims and experiments stated in the main text.

## A.1    DISCUSSING THE OPTIN ALGORITHIM

Algorithm 1 demonstrates the base structure for computing and assigning importance to each of our printable parameters. Once our importance scores were computed we directly leveraged the mask search algorithm from PTF (Kwon et al., 2022) as it searches to maximize scores in the partitioned search space. Their policy partitions the search space by incrementally adding attention heads in order of importance, and at each step, adding the maximum number of rank-ordered neurons that will satisfy the cost constraint $\mathcal{C}([\theta_0', \theta_1', \cdots \theta_n']) \leq C$. By computing the cumulative importance at each step, we easily deduce that the step with the maximum cumulative importance must be optimal, as any other configuration would yield a cumulative score less than or equal to that of the best.

## A.2    DETAILS ON HYPERPARAMTER $\lambda$

In Tab 1d, we use the $\lambda$ sweep to express relative magnitude differences between $\mathcal{L}_{MD}$ and $\mathcal{L}_{KD}$. Based on the reported results, we can conclude that a larger importance should be weighed on the distillation loss, in particular, we expect that $\lfloor \log_{10}(\mathcal{L}_{MD}) \rfloor \sim \{10, 100\} * \lfloor \log_{10}(\mathcal{L}_{KD}) \rfloor$ (i.e. the order of magnitude of $\mathcal{L}_{MD}$ should be 10-100 times larger than that of $\mathcal{L}_{KD}$)

## A.3    DETAILS ON DATASET CHOICE

In this work, we evaluated the strength of the OPTIN framework on natural language processing benchmarks, image classification and semantic segmentation. Across this wide variety of domains, we used different datasets for the various tasks following from previous works to bolster the performance of our method.

**GLUE Benchmarks** The General Language Understanding Evaluation (GLUE) benchmark (Wang et al., 2019) contains a variety of NLP-related tasks of which we included: Similarity and Paraphrase Tasks (MRPC STS-B, QQP) and Inference Tasks (MNLI, QNLI). The dataset distribution widely vary per task. From the included selection, STS-B is a regression task, MNLI has three classes, and the remaining tasks include two classes.

**ImageNet 1K Benchmarks** The ImageNet-1K dataset (Deng et al., 2009) is a widely used benchmark for image classification, as cited in several works (Zhu et al., 2021; Goyal et al., 2020; Pan et al., 2021). It contains 1.2 million training images and 50K validation images across the 1000 classes. Due to its difficulty, stemming from the number of classes and images, it presents a perfect benchmarking medium for our one-shot pruning approach.

**CIFAR 10** The CIFAR-10(Krizhevsky et al., 2009) datasets are a common benchmark across both convolutional neural network pruning (Khaki & Luo, 2023; Lin et al., 2020a) CIFAR10 contains 50K training and 10K validation images spread over 10 classes. Due to the prominent use of this dataset in benchmarking tasks, we decided to benchmark our method for fair comparison with SOTA.

**Cityscapes** The Cityscapes dataset (Cordts et al., 2016) is heavily used for semantic segmentation tasks, and in particular, contains high-resolution images of (1024x2048), with roughly 3K training and 500 validation images. In this paper, our goal was to demonstrate the effects of pruning a downstream network under the OPTIN framework, and due to the high resolution of Cityscapes data, we were able to demonstrate improvements in throughput speed for our pruned segmentation model.

## A.4    DETAILS ON EVALUATION METRIC CHOICE

The two main metrics reported in this work are FLOP(s) and Accuracy (or equivalently mIOU in the case of segmentation). Given the target domain of this paper, these metrics best express the tradeoff between high accuracy and computational complexity, and further how the OPTIN framework is better able to make this distinction. The selected metrics have further been reported in similar

previous works (Kwon et al., 2022; Bolya et al., 2023; Wei et al., 2023). Finally, we also report im/s throughput to better illustrate the real-world implications of the OPTIN Framework, especially in resource or time-constrained environments.

## A.5 AVERAGE TIME ANALYSIS

| Task | Dataset | Model | Avg. Pruning Time (Hours) |
|---|---|---|---|
| Natural Language | GLUE Benchmark | BERT | 0.4 |
| Image Classification | ImageNet-1K | DeiT Tiny | 0.3 |
| Image Classification | ImageNet-1K | DeiT Small | 0.3 |
| Semantic Segmentation | Cityscapes | Mask2Former(Swin-Ti) | 0.5 |

## A.6 ABLATIVE EXPERIMENTS

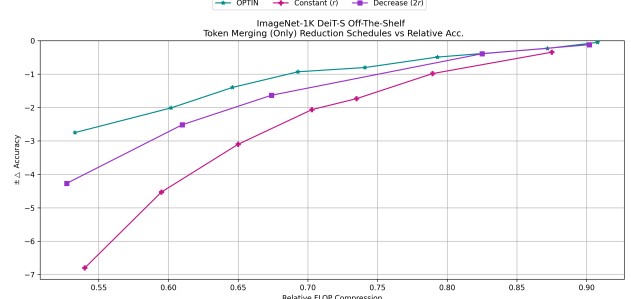

| Dataset | Batch Size | Acc. |
|---|---|---|
| MNLI | 16 | **82.12** |
| MNLI | 32 | **81.90** |
| MNLI | 64 | 81.73 |
| MNLI | 128 | **82.20** |
| ImageNet | 16 | 70.53 |
| ImageNet | 32 | **71.25** |
| ImageNet | 64 | 71.01 |
| ImageNet | 128 | 70.82 |

(a) **Effect of Batch Size.** The computation of $\mathcal{I}$ appears to be more robust to changes in the batch-size.

(b) **Ablating the Token Reduction Schedule.** On ImageNet-1K for DeiT-S, our $\text{OPTIN}_{\tau(\infty)}$ Framework produced a more optimal reduction scheme as opposed to the default in ToMe (Bolya et al., 2023)

| Token Merging Alg. | Scheduler | $\pm \triangle Acc(\%)$ | FLOPs(G) |
|---|---|---|---|
| Baseline (N/A) | - | - | 17.6 |
| Random Prune | default | ↓ 8.47 | 11.81 |
| Random Prune | $\text{OPTIN}_{\tau(\infty)}$ | ↓ **7.11** | **11.75** |
| bipartite merge | default | ↓ 0.80 | 11.81 |
| bipartite merge | $\text{OPTIN}_{\tau(\infty)}$ | ↓ **0.52** | **11.75** |

(c) **Ablating the Token Merging Algorithm.** On ImageNet-1K for DeiT-B, the bipartite matching default configuration from ToMe was used. However based on the results, it is evident that OPTIN improves the performance of arbitrary token pruning methods as well. Configuration Ablated: $\text{OPTIN}_{\tau(\infty)}$

Table 6: **Additional Experiments** Here we evaluate three components: (a) the ablative effect of batch size in computing the distillation loss, (b) a comparison of the optimal reduction schedule from $\text{OPTIN}_{\tau(\infty)}$ compared to the constant and decreasing schedules from ToMe. (c) the effect of different patch reduction/merging techniques using the OPTIN Framework. Our default settings are marked in green .

| Method | MNLI | QQP | QNLI | SST | STS-B | MRPC |
|---|---|---|---|---|---|---|
| $\text{BERT}_{BASE}$ | 84.53 | 91.00 | 91.41 | 93.57 | 88.90 | 86.27 |
| PTF | 81.21 | 89.99 | 88.38 | 92.13 | 87.10 | 83.14 |
| $\textbf{OPTIN}^{\ddagger}_{\lambda_c=0.01}$ | **82.12** | **90.08** | **88.54** | **92.36** | **87.19** | **85.21** |
| $\text{PTF}^{\dagger\dagger}$ | 82.51 | 90.35 | 90.06 | 92.49 | 88.00 | 85.27 |
| $\textbf{OPTIN}^{\ddagger\ddagger}$ | **82.74** | **90.43** | **90.35** | **92.73** | **88.21** | **85.68** |

Table 7: **Natural Language Benchmarks.** Augments the main table 7 with two additional experiments. $\textbf{OPTIN}^{\ddagger}_{\lambda_c=0.01}$ runs the OPTIN algorithm with $\lambda_c = 0.01$ to display the best results we achieved using our standard framework. Further, $\textbf{OPTIN}^{\ddagger\ddagger}$ compares with $\text{PTF}^{\dagger\dagger}$ which includes the mask tuning/scaling from PTF (Kwon et al., 2022) to discover a non-binary mask that helps to reduce in-place reconstruction errors. Latency is estimated at $B = 32$ and ranges between 1.35-1.38 $\times$ improvement.$^{\ddagger}$ results are averaged over 5 different seeds.

| Task | Attention Heads | Hidden Neurons | Patches&Tokens | Output Channels |
|---|---|---|---|---|
| Natural Language Processing | ✓ | ✓ | - | - |
| Image Classification (CNN) | - | - | - | ✓ |
| Image Classification (TF) | ✓ | ✓ | ✓ | - |
| Semantic Segmentation | - | ✓ | - | - |

Table 8: Identifying the prunable weights that OPTIN uses to accelerate the model for various downstream tasks

## A.8 EXTENDING OPTIN TO VARIOUS DOWNSTREAM TASKS

When moving from the language domain to other applications, competitive methods leverage additional pruning components in order to spread the compression over a larger search space. In response, we too apply this with OPTIN. Tab 8 identifies the search space used in OPTIN for various downstream tasks.

## A.9 ADAPTING THE TRAJECTORY FORMULATION TO TOKENPATCH INFORMATIVENESS

As detailed in Sec 4.2, the OPTIN framework allows users to select the best token reduction technique for their task to create an expanded prunable search space. The OPTIN framework adapts to image datasets by further producing an optimal reduction schedule for tokens that can be leveraged by any reduction or merging technique. In the main paper, we use ToMe with bipartite matching, however, we ablate the metric choice and merging strategy in Appendix A.6.

In order to obtain the optimal token reduction, we apply the trajectory estimation to patches in the vison-transformer models, by simply modifying the reshaping operator and the dimension upon which we compute the importance. We adopt the inter-sample representation (Hao et al., 2022) and since we are determining patch-level importance, it follows that we should compare our base and pruned embeddings along said dimension. We note that the use of the term *patches* in the context of vision-transformers would be represented by the same dimension as the token sequence length in the language domain. We redefine the manifold distillation loss according to the index $j$ which ranges up to the number of patches for the co-responding model. We begin by redefining the manifold structure relational map based on index $j$ where $F_{i,[:,j,:]} \in \mathbf{R}^{B \times 1 \times D}$ as:

$$\mathcal{M}(F_{i,[:,j,:]}) = (F_{i,[:,j,:]})(F_{i,[:,j,:]})^T$$

In particular, we modify the $\mathcal{L}$ inter-image patch distillation loss from (Hao et al., 2022) by replacing the student input with that of the masked patch, and the teacher input with the precomuted embeddings from the network. For two given feature embeddings from layer $i$ for a base and pruned model, the

sample manifold reconstruction error would present as:

$$\mathcal{L}_{MD}(F_i^{'}, F_i) = \frac{1}{T} \sum_{j=0}^{T} ||\mathcal{M}(F_{i,[:,j,:]}^{'}) - \mathcal{M}(F_{i,[:,j,:]})||_F^2$$

This error is accumulated with standard KL Divergence resulting in a similar Equation 3.

After determining the importance score for each patch, we can derive the average number of patches required per layer for maximum information throughput by simple rank elimination given an FLOPs constraint. After executing the mask search with attention heads and neurons, we alleviate the remaining FLOP reduction required by removing tokens in order of ascending importance (i.e remove the lowest importance first). This ultimately produces the number of tokens per layer which can be extracted as a token reduction schedule that can be leveraged on run-time with the ToMe bipartite matching scheme. We have further shown that the OPTIN Framework reduction scheme is much more informed than the standard constant or decreasing schemes commonly used – See Appendix A.6.

### A.10  ADAPTING THE TRAJECTORY FORMULATION TO OUTPUT CHANNELS

To adapt the trajectory estimation to output channels in CNN-style networks, we reduce the relation to a simple mean-square error computed between the feature embeddings along the length of the model. In particular, we average the embeddings along the batch dimension and compute the sum of the mean squared error between the base and pruned model along each layer deeper in the network:

$$\mathcal{L}_{MD}(F_i^{'}, F_i) = \frac{1}{B} \sum ||\sum^{B} F_i^{'} - \sum^{B} F_i||_F^2$$

Once again, we are able to plug this into Equation 3 with standard KL Divergence to determine overall channel importance.

