# OpenReview forum: "The Need for Speed: Pruning Transformers with One Recipe"
_ICLR.cc/2024/Conference — ICLR 2024 poster_

### Official Review · Reviewer_meSa · 2023-10-14

**Soundness:** 3 good
**Presentation:** 3 good
**Contribution:** 3 good
**Rating:** 6
**Confidence:** 3

**Summary:**

The authors propose a post-training pruning method that operates iteratively. The idea is to approximate each weight's impact on the validation error. To achieve this, the authors mask out weights, then run the forward pass and compute the change in logits.

**Strengths:**

- The paper presents a significant number of experimental results and ablations.  The ablations in Table 1 are convincing evidence that the design choices made were fully vetted.
- Overall, the study seems very thorough. I don't have any strong reasons to reject or to accept. (suggestions below)

**Weaknesses:**

- It seems like PTF is the most relevant and competitive baseline. Per 4.1, the authors compare with an abridged version of PTF however. If you use the same tricks that PTF used (e.g., mask rearrangement -- maybe mask tuning), would your method outperform PTF?
- Figure 1 is a bit complicated to look at. One possibility is to break up this figure into two (the left vs. the right). The details are nice, but they detract from the main point (e.g., the activation difference and the final difference in logit distribution). The logits could be illustrated as two histograms, highlighting their difference.
- The double axis in Figure 2 right is really confusing. Maybe just add a 5th plot? Might not be so easy to see for anyone with color blindness.
- How long does the search take to run? On the order of minutes? Hours? Days? PFT said it took minutes, so that seems like the baseline. Is there significant latency overhead in storing and fetching large activations in RAM, especially for vision models?
- The paper's motivation could be stronger. The abstract and title focus heavily on "without retraining" but it seems like many previous papers already focus on the post-training regime (as the paper cites). The more important motivation is then: What is PFT lacking that is present in this method? *Why is that difference important, intuitively?


Summary: In short, I'm very lackluster about the paper. The results seem reasonable, but the motivation and the presentation (figures for example) can be improved to make a more convincing case.

**Questions:**

- typo "Perforamnce" in 4.1
- Every instance of "without re-training" is italicized; it's a bit excessive to do that, particularly given there is previous work that already introduced the idea.

---

> ### Author Response · Authors · 2023-11-20
> **Response to Reviewer meSa (1/2)**
>
> We appreciate reviewer meSa’s recognition of our comprehensive experiments and well-vetted design choices. We provide responses to the questions below in an effort to bring more clarity to our work:
>
> $\textbf{[Applying PTF Tricks]}$
> In the language domain, PTF presents the most competitive baseline. We are augmenting our results using the associated tricks from PTF and have provided them in the below table. Improvement tricks, such as those applied here, often boost performance in narrow applications and generally require data or time that a practitioner may find difficult to execute. For these reasons, we didn’t initially report these augmented results; our key motivation, in this paper, is to show generality over a variety of tasks and domains.
>
> | Method      | MNLI | QQP | QNLI | SST | STS-B | MRPC |
> | ----------- | ----------- |----------- |----------- |----------- |----------- |----------- |
> | $PTF_{+tricks}$      | 82.51 | 90.35 | 90.06 | 92.49 | 88.00 | 85.27 |
> |$OPTIN_{+tricks}$   | $\textbf{82.74}$ | $\textbf{90.43}$ | $\textbf{90.35}$ |$\textbf{92.73}$ | $\textbf{88.21}$ | $\textbf{85.68}$ |
>
>
> $\textbf{[Improving Figures]}$
> We appreciate the reviewer’s comments on Figure 1. We have made another attempt to separate the details from the framework diagram to develop a more pleasing and easy-to-understand image. We also greatly appreciate your comment on our Figure 2. We had made every effort to add an additional distinguishing feature for different methods, (i.e. marker shape + color in each plot). We had missed this one in our review, and are updating it. We also split the plot into two, as we too can understand the difficulty posed by the overlay. We will reply in this thread once the main paper has been updated to reflect this change.
>
> $\textbf{[Pruning Time/Memory]}$
> We thank the reviewer for this question. This helps elucidate one of our main contributions. As such, we have attached a table below detailing the time taken to prune various models on different datasets. We’d like to reiterate that our method appears to scale well across a variety of architectures and tasks, achieving competitive performance-to-compression ratios on the scale of minutes. There is some overhead time required in computing the activations however this is included in our pruning time listed below. In terms of memory allocation, all our pruning experiments, across both language, and vision (including semantic segmentation), can be run with 1x Nvidia RTX2080 with less than 10 GB of memory.
>
> $\textbf{Average Time Analysis for } OPTIN$
> | Task  | Dataset  | Model      | Pruning Time (Hours) |
> | ----------- | ----------- |----------- | ----------- |
> |Natural Language| GLUE Benchmark | BERT | 0.4 |
> |Image Classification| ImageNet-1K | DeiT Tiny | 0.3 |
> |Image Classification| ImageNet-1K | DeiT Small | 0.3 |
> |Semantic Segmentation| Cityscapes | Mask2Former(Swin-Tiny) | 0.5 |
> $\textit{*Our method does not incur re-training time.}$

---

> > ### Author Response · Authors · 2023-11-22
> > **Response to Reviewer meSa (2/2)**
> >
> > $\textbf{[Motivation + Italics]}$
> > We appreciate the reviewer’s comments on motivation. The practical applications of our methods is central to the paper’s design, and it is thus crucial that a reader understands the novelty and applicability of our method. Restating our contributions, in reference to PTF:
> >
> > * [C1] Our OPTIN framework leverages the trajectory measure, which efficiently captures long-range dependencies between parameters in the network. PTF approximates the fisher-information matrix to determine the relative importance of each parameter in the network. However, this method of ranking parameters is highly subject to parameters that may exhibit high gradients and thus fails to sufficiently capture relationships between parameters across layers. In comparison, our method uses the forward pass to establish and capture relationships between successive parameters ultimately achieving superior performance.
> >
> > * [C2] The OPTIN framework has exhaustively verified superior performance across language and vision domains, including an extension to CNN architectures and segmentation models. Meanwhile, PTF has only shown applicability in the language domain. This provides significant motivation for our method, as it can be easily adopted as a plug-and-play across various domains in deep learning. We would really like to highlight this generalizability to multiple domains and model architectures, as many works depend on the intricacies of a particular type of data or model in order to prune efficiently.
> >
> > We are in the process of refreshing our Abstract, Introduction, and Related Works to best elicit this motivation, primarily focusing on the multi-modality and widescale applicability of our pruning framework.
> >
> > Finally, we thank the reviewer for noticing the italicized words. This was most definitely a formatting error/typo on our side. We meant to italicize only once in the abstract and once in the introduction, and are correcting this in the updated draft.
> >
> > We hope these responses have helped clarify any questions the reviewer may have. We are adjusting the main paper to better focus on said motivation as well as improve the quality and presentation of our figures. We hope you will consider our changes and improvements in order to best evaluate the potential of our work. Please feel free to reach out with any questions, we’d be more than happy to follow up with additional improvements or changes.

---

> > > ### Comment · Reviewer_meSa · 2023-12-03
> > > **Thank you for the clarifications, bumped up**
> > >
> > > Thanks to the authors for the clarifications, additional results, and additional statistics. This addresses my outstanding concerns, so I've bumped my score up.
> > >
> > > - The algorithm runtime was longer than I expected, but less than an hour is enough.
> > > - Glad that OPTIN outperforms PTF in all scenarios for the table above, after "normalizing" by tricks used.
> > > - The suggested/effected changes to the manuscript also sound good.

---

### Official Review · Reviewer_oBzK · 2023-10-23

**Soundness:** 3 good
**Presentation:** 3 good
**Contribution:** 3 good
**Rating:** 6
**Confidence:** 2

**Summary:**

This paper introduces One-shot Pruning Technique for Interchangeable Networks (OPTIN) to prune networks with re-training. With intermediate knowledge distillation, the proposed method saves remarkable computational costs. Experiment results prove the effectiveness of the method.

**Strengths:**

1. This paper targets on a practical problem, model compression in a more efficient way.
2. Overall this paper is well-written and easy to follow. It presents the proposed method very clearly.

**Weaknesses:**

1. I think this paper is a ok paper. For discussion, I want to see the method performance on more complicated network architecture or datasets.
2. I shall mention that I am not very familiar with the baseline methods. Therefore, if the other reviewers points out the missing related works, please add the comparison results.

**Questions:**

See weakness.

---

> ### Author Response · Authors · 2023-11-20
> **Response to Reviewer oBzK**
>
> We thank reviewer oBzK for recognizing the practical problem we are aiming to solve and appreciate the reviewer’s comments on the clarity and comprehensibility of our methodology.
>
> $\textbf{[More Architectures or Datasets]}$Following the suggestions for more experimentation, we have extended our method to the task of language generation using GPT-2 to show versatility (see below). We would however like to reiterate the diversity of tasks and modalities already included in the main paper. We were able to show competitive performance in both the natural language and image classification benchmarks. We further demonstrated applicability to tasks including transfer learning, and the high-resolution semantic segmentation. We highlight the complexity of semantic segmentation as the practical improvements in this domain could lead to huge potential impacts, particularly in the field of autonomous driving. Semantic segmentation on the cityscapes datasets operates at a resolution of 1024x2048, which is significantly higher than common resolutions for classification, making inference speed a huge bottleneck. By showing applicability to this style of model, we open avenues for future research to investigate the multi-modal capability of efficient model compression algorithms. Further, we demonstrate the robustness of our OPTIN framework as it is able to achieve competitive compression ratios on a variety of tasks. We would like to highlight that our main contribution is the generality and versatility of our one-shot approach; Some methods exhibit tricks or artifacts to improve performance in specific applications, however, our design approaches the problem from a more general perspective of a model compression framework that scales well across modalities.
>
> $\textbf{Extending OPTIN to Language Generation}$
> To show versatility, we produce an initial experiment on Language Generation. Our framework has primarily focussed on model compression techniques for transformers. Given their prevalence in modern deep learning, the research into efficient deep learning attempts to solve a practical problem across the industry. In particular, we’ve shown the flexibility and extensibility of our method over a variety of tasks. In this section, we produce some initial findings by extending our framework to language generation tasks. In particular, we evaluate the GPT-2 model on the WikiText2, PTB, and LAMBDA datasets. Unlike our previous results in the main paper, these datasets are used to evaluate a model's ability in language generation. This differs from the standard GLUE benchmark in NLP that mainly measures accuracy/correlation. We hope to extend our tests in the main paper, as showing generalizability of our method to a primarily decoder-related task, such as language generation, opens multiple avenues for future extensions, including those in image-generation.
>
> | Ratio (%)  | Model | Pruning Time (Hours) | ~ $\pm \triangle WikiText2$ (PPL)  | ~ $\pm \triangle PTB_{Text} $ (PPL)  | ~ $\pm \triangle LAMBDA_{Text} $ (PPL)  |
> | ----------- | ----------- |----------- | ----------- | ----------- |----------- |
> | 1.0 | GPT-2| -|(28.2) | (31.9) | (42.3) |
> | 0.9  | GPT-2 | 0.3 | +3.8|+4.1|+2.5|
> | 0.8  | GPT-2 | 0.3 | +4.1|+6.3|+5.1|
>
> $\textit{* First row includes the approximate baseline for each dataset.}$
> $\textit{* Pruning time is insured only once. Activations are stored and re-used for various compression ratios.}$
> $\textit{* Relative Parameter Ratio accounts for the hidden dimension and attention heads.}$
> $\textit{* PPL referenced Perplexity, a lower measure is better for language generation}$
>
> We hope that our main results on Semantic Segmentation, and the additional introductory experiments here provide sufficient inclination towards the future potential impact of such a generalizable model compression framework. We believe that there is strong potential for future avenues of research that can stem from achieving a generalizable approach to compressing transformer models across modalities, and we hope that our work compellingly conveys the message. We would like to thank the reviewer again for their time and interest in our work.

---

### Official Review · Reviewer_RsPn · 2023-10-30

**Soundness:** 3 good
**Presentation:** 2 fair
**Contribution:** 2 fair
**Rating:** 6
**Confidence:** 3

**Summary:**

The paper introduces OPTIN, a framework designed to prune neural networks without the necessity of additional training. It prunes weights that have least impact on the intermediate embeddings and final logits within neural networks. The authors have tested OPTIN across a variety of tasks in both vision and language domains, showcasing its potential effectiveness.

**Strengths:**

1. **Comprehensive Experiments**: The authors have conducted a wide array of experiments, covering domains such as vision, language, and semantic segmentation tasks, ensuring a comprehensive evaluation of OPTIN.
2. **Strong Performance**: OPTIN demonstrates robust results across the tasks it was tested on, showcasing its effectiveness in network pruning.

**Weaknesses:**

1. **Performance on Vision Models**: As highlighted in Table 3, the performance of OPTIN (without token reduction) appears to be inferior to that of VTP and PoWER for both evaluated models. It seems that the impressive performance of $OPTIN_\tau$ can be mainly attributed to token reduction. Does it mean OPTIN is more suitable for language models rather than vision models?

Minor issues:
1. Table 1: "(a)Explores" -> "(a) explores".
2. A.6: "Tab ??"

**Questions:**

1. In Table 2, the results for PTF are directly cited from the original paper. Could the authors confirm that the pre-trained checkpoints and fine-tuning settings are consistent across all experiments to ensure a fair comparison?
2. Could the authors provide details on the estimated time required for OPTIN to prune different models across various datasets? This information would be helpful to evaluate the practicality of implementing OPTIN in different scenarios.
3. Could the authors give further explanations on 'IM-Dense (After dense embedding layer)'. What does it mean?
4. How does OPTIN determines layer-wise sparsity? Does OPTIN rank weights globally?
5. In Table 1.(d), why $\lambda_c$ is chosen to be 0.1 even though 0.01 provides better results?

I am willing to adjust my rating if my questions are addressed.

---

> ### Author Response · Authors · 2023-11-20
> **Response to Reviewer RsPn (1/2)**
>
> We thank reviewer RsPn for their thoughtful and insightful comments about our paper. We appreciate their recognition of our comprehensive experiments and robust performance. We hope to provide more insight on their questions below:
>
> $\textbf{[Table 2 Results and Checkpointing]}$
> In Table 2, we use the checkpoints provided by the authors of PTF to replicate the baseline performance. Applying the OPTIN method to the baseline model incurs no fine-tuning or additional training costs as our method masks the least important parameters, and thus we can ensure both methods are compared fairly as they start with the same model.
>
> $\textbf{[Time Requirement]}$
> We agree with the reviewer that a time comparison can help validate the practicality of our method and showcase the true time benefit of an approach that is one shot. As such, we have added a table below to showcase the time costs of our method. This table shows the practicality of our method as it scales well across the tasks and model architectures, achieving competitive results on the scale of minutes. If successful, we hope to integrate this into our main draft to better depict the motivation and implications of our work.
>
> $\textbf{Average Time Analysis for } OPTIN$
> | Task  | Dataset  | Model      | Pruning Time (Hours) |
> | ----------- | ----------- |----------- | ----------- |
> |Natural Language| GLUE Benchmark | BERT | 0.4 |
> |Image Classification| ImageNet-1K | DeiT Tiny | 0.3 |
> |Image Classification| ImageNet-1K | DeiT Small | 0.3 |
> |Semantic Segmentation| Cityscapes | Mask2Former(Swin-Tiny) | 0.5 |
> $\textit{*Our method does not incur re-training time.}$
>
> $\textbf{[Ablation Experiment in Table 1a - IM Dense]}$
> The purpose of this ablation is to decide at which location to extract the features used in our distillation loss. Following recent works, including [1,2], we noted that there are multiple locations from which we can extract meaningful features for our distillation loss. At the end of each transformer block there exists a 2-layer MLP (in the case of BERT models). The first layer increases the hidden dimension, while the second is complimentary in operation. In this ablative study, we compare the location to compute distillation error in subsequent transformer blocks. FFN refers to extracting after the 1st  linear layer at the increased embedding dimension, thus leveraging a more robust feature comparison, while IM-Dense refers to extracting after the 2nd linear layer. Our study ultimately concluded that a higher degree of information regarding parameter importance could be extracted from the distillation error computed along the larger hidden embedding dimension. We will revise the in-text terminology to make this clearer.
>
> *[1] Peng et al. 2019: Correlation Congruence for Knowledge Distillation
> *[2] Hao et al. 2022: [see main paper]
>
> $\textbf{[Sparsity]}$
> Previous works have shown the benefits of using a global sparsity as opposed to layerwise sparsity [3,4]. In line with these works, we too exploit a global sparsity constraint as we rank the importance of each parameter globally in the network. See Details in Appendix A.1.
>
> *[3] Lin et al. 2018: Accelerating Convolutional Networks via Global dynamic Filter Pruning
> *[4] Song et al. 2022: [see main paper]
>
> $\textbf{[Lambda Value]}$
> We appreciate the reviewer pointing this out. The goal of our ablations was to test the sensitivity to different parameters and validate our design choices. In the early inception of our project, we defaulted to using 0.1 for language models, to favor the trajectory measurement in computing parameter importance. We hoped this ablative study could help elucidate the importance of trajectory, however, we note that the values of our main language table could be improved with this lambda parameter, and have rerun the language experiments with the updated lambda value. As can be seen in the table below, there is an improvement across most tasks in the GLUE benchmark when compared to the previous configuration.
>
> | Method      | MNLI | QQP | QNLI | SST | STS-B | MRPC |
> | ----------- | ----------- |----------- |----------- |----------- |----------- |----------- |
> | $OPTIN_{\lambda=0.1}$      | 81.90 | 90.06  | 88.49 | 92.24 | $\textbf{87.25}$ | 85.13|
> | $OPTIN_{\lambda=0.01}$   | $\textbf{82.12}$ |$\textbf{90.08}$ | $\textbf{88.54}$ | $\textbf{92.36}$ |87.19 |  $\textbf{85.21}$|

---

> ### Author Response · Authors · 2023-11-20
> **Response to Reviewer RsPn (2/2)**
>
> $\textbf{[Addressing Weaknesses]}$
> We very much appreciate the time taken by reviewer RsPn to understand our work. With regard to the competitive methods in the vision domain, it seems that each method has its own approach to the granularity of pruning. In particular, VTP performs layer-wise patch-wise pruning while also removing parameters in the QKV and linear layers within each transformer block. PoWER introduces a soft-extract layer in each self-attention module to determine which tokens are kept at each self-attention layer and CP-ViT performs head and patch pruning under a dynamic search. The focus on patch-based reductions enables these methods to achieve larger FLOP reduction rates with minimal parameter changes. Each method exhibits its own artifacts which make it hard to compare fairly. In our case, we justify the inclusion of token reduction (sometimes referred to as patch reduction), to achieve a more fair comparison across the methods. Our base $\beta$ configuration directly removes parameters, while the augmented $\tau$ configuration leverages token reduction to further achieve more competitive flops reduction ratios. Further, we’d like to reiterate that even under token reduction techniques, our method still outperforms state-of-the-art approaches, as seen in Table  4. For these reasons, we find our method to be applicable in the vision domain.
>
> We hope that these answers have helped to clarify any questions you may have about our work. Please feel free to ask any follow-up questions you may have to help us best exemplify our work.

---

> > ### Comment · Reviewer_RsPn · 2023-11-22
> > **Thank you for your reply.**
> >
> > Thank you for your clarifications. Most of my concerns are addressed and I will keep my rating.

---

> > > ### Author Response · Authors · 2023-11-23
> > > **Thank you to Reviewer RsPn**
> > >
> > > We would like to thank Reviewer RsPn for confirming that their concerns were addressed. We would like to ask if the reviewer has any other concerns that we could help address in order to best evaluate our work.

---

### Author Response · Authors · 2023-11-20
**Reviewer Appreciation**

Foremost, we would like to thank all the reviewers for their invaluable contributions to our work, in the way of insightful feedback and diligent assessments that can be leveraged to greatly enhance the quality and impact of our work. We have responded to each reviewer individually for their respective questions and will be following up in the threads as more experimental results become available.

---

### Meta-Review · Area_Chair_dNni · 2023-12-05

**Metareview:**

This paper introduces a simple framework for pruning transformers that does not require additional training. The central idea is to prune weights that have the least impact on the intermediate embeddings and the final logits of the network. The pruning framework is evaluated on both vision and language models.
The experiments in this paper are quite comprehensive showing that the technique generalizes well. The domain-specific techniques (like PTF) still outperform the proposed method albeit trading off complexity and domain-specific tricks.

**Justification For Why Not Higher Score:**

The paper has good performance across both vision and language domains. However, there is still a gap with domain specific pruning techniques. It is unclear how this gap scales with model size.

**Justification For Why Not Lower Score:**

The paper has a simple insight for pruning that does not require training. It has good results on multiple different benchmarks.

---

### Decision · Program_Chairs · 2024-01-16

Accept (poster)